

**Optimized fertilization using online soil nitrate data**
Yonatan Yekutiel[1], Yuval Rotem[1], Shlomi Arnon[2], Ofer Dahan[1]
[1]Department of Hydrology & Microbiology, Zuckerberg Institute for Water Research, Blaustein Institutes
for Desert Research, Ben-Gurion University of the Negev, Israel
[2]Electrical and Computer Engineering Department, Ben-Gurion University of the Negev, Israel
*Correspondence to*: Yonatan Yekutiel (yekuyona@post.bgu.ac.il)
## Abstract
Managing fertilizer application according to actual soil nutrient availability is a key
strategy for achieving sustainable agriculture and a healthy environment. A new soil
nitrate monitoring system that was installed in cultivated field enabled, for the first time,
controlling the nitrate concentration across the soil profile. The monitoring system was
installed in a full-scale agricultural greenhouse setup that was used for growing a bell
pepper crop. Continuous measurements of soil nitrate concentrations were performed
across the soil profile of two plots: (a) an experimental plot, in which the fertigation
regime was frequently adjusted, according to the dynamic variations in soil nitrate
concentration and (b) a control plot, in which the fertigation was managed according
to a predetermined fertigation schedule that is standard practice for the area.
The results enabled an hourly resolution in tracking the dynamic soil nitrate
concentration variations, in response to daily fertigation and crop demand. Nitrate
concentrations, in and below the root zone, under the control plot, reached very high
levels of ~800 ppm throughout the entire season. Obviously, this concentration reflects
excessive fertigation, which is far beyond the plant demand, entailing severe
groundwater pollution potential. On the other hand, frequent adjustments of the
fertigation regime, which were carried out under the experimental plot, enabled control
of the soil nitrate concentration around the desired concentration threshold. This
enabled a dramatic reduction of 38% in fertilizer application, while maintaining
maximum crop yield and quality. Throughout this experiment, decision-making on the
fertigation adjustments was done manually based on visual inspections of the soil's
reactions to changes in the fertigation regime. Nevertheless, it is obvious that an
algorithm that continuously processes the soil nitrate concentration across the soil
profile and provides direct fertigation commands could act as a "fertistat" that sets the



soil nutrients at a desired optimal level. Consequently, it is concluded that fertigation that is based on continuous monitoring of the soil nitrate concentration may ensure nutrient application that accounts for plant demand, improves agricultural profitability, reduces nitrate down-leaching, and eliminates water resource pollution.

## 1. Introduction

Groundwater pollution by nitrate constitutes one of the main factors in freshwater disqualification worldwide (Li et al., 2021; Abascal et al., 2022). High nitrate levels in drinking water have been correlated to health issues, such as digestive tract cancer (Powlson et al., 2008; Picetti et al., 2022) and blue baby syndrome (Wisconsin et al., 2000). In addition, excessive nitrate in the environment leads to algal blooms that, in turn, cause eutrophication and hypoxia in surface water bodies, such as rivers, lakes, and even oceans (Bijay-Singh and Craswell, 2021; Górski et al., 2019; Zhang et al., 2021; Wang et al., 2018; Scavia and Bricker, 2006). Water resource pollution by nitrate is primarily attributed to intensive agricultural fertilizer application (Li et al., 2023; Rahmati et al., 2015; Gu et al., 2013; Zhang et al., 2014). Excessive fertilization results in the down-leaching of nitrate from the soil through the unsaturated zone to the groundwater. Ultimately, polluted groundwater naturally discharges to associated surface water bodies (Lasagna et al., 2016) or is pumped from abstraction wells for direct use.

Nitrogen-use efficiency in agriculture refers to the fraction of nitrogen from the applied N-fertilizer that is consumed by the plant. Unfortunately, on a global scale, nitrogen-use efficiency is very low, with an estimated worldwide average of 55% (FAO, 2022). Accordingly, any attempt to control nitrogen pollution in water resources requires fertilizer management that follows the actual dynamics of crop nitrogen demand, avoids excess fertilization, and prevents nitrate leaching from the soil to the groundwater. This currently constitutes one of the greatest environmental challenges and a critical milestone for sustainable agriculture.

Presently, agricultural fertilization management relies mainly on predetermined programs that are based on farmers' experience, expert knowledge, and fertilizer manufacturers' recommendations, all of which primarily aim at maximizing crop yield. In practice, none of these fertilization practices correspond well with the actual



dynamics of fertilizer mobility in the soil and plant uptake. Hence, most commonly, fertilization programs are deliberately designed for excess fertilization to prevent potential nutrient deficiencies and yield reduction. As a result, a major portion of the N-fertilizers ends up as nitrate, which is either transported with the irrigation water below the root zone to the groundwater or transformed into N-oxides, which may be released into the atmosphere (Minikaev et al., 2021). Unfortunately, the lack of real-time information on the nutrient concentration in the soil during the growing season pushes farmers to undesirable excessive fertilization, regardless of the devastating environmental consequences.

Traditionally, controlled agricultural experiments have provided the basis for all agricultural development and fertilization protocols (Salvagiotti et al., 2008; D. T. Westermann and G. E. Kleinkopf, 1985). Obviously, these experiments primarily aim at achieving the highest yield, while increasing nitrogen efficiency and reducing the overall costs of agricultural inputs (Cui et al., 2008; Li et al., 2007; Lollato et al., 2019; Piri and Naserin, 2020; Kurtzman et al., 2021; Nkebiwe et al., 2016). Such experiments naturally span over time scales of years, and implementing their results in the agricultural industry may take much longer. Along with field agricultural experiments, optimizing fertilization regimes is often investigated through numerical simulations, which are validated using data from field and controlled experiments (Zhang et al., 2020; Rezayati et al., 2020; Tafteh and Sepaskhah, 2012; Azad et al., 2019; Xu et al., 2020; Sela et al., 2018). Despite their robustness, numerical simulations require many variables, which depend on crop type, soil properties, atmospheric conditions, plant uptake, etc. (Šimůnek et al., 2016). Unfortunately, these variables are often very vague, with a wide range of spatial and temporal variability, which reduces the applicability for large-scale or variable environmental conditions (Weissman et al., 2022). Moreover, these methodologies do not provide a real-time response to the temporal variation in soil nutrient conditions, which often results in over-fertilization.

Fertilization adjustment during the growing season is often based on measurements of the plant's state. These are often carried out through either tissue analysis, such as chlorophyll content (Bijay-Singh and Ali, 2020; Mohamed ALI et al., 2015), or leaf spectral measurement (Bijay-Singh et al., 2015; Feng et al., 2008). These methods provide important indications of plant "health." Nevertheless, due to the time lag in the



plant's natural response to the soil nutrient state, observable phenotypic changes
provide late indications of nutrient problems in the soil. Moreover, these methods can
only detect nutrient deficiency and are not effective in detecting nutrient excess, which
is a key factor in reducing environmental pollution.
Soil nutrient content is commonly determined through either water extraction from soil
samples or analysis of soil porewater obtained by suction cups (suction lysimeters)
(Carter and Gregorich, 2007). These water samples are then analyzed through
standard laboratory techniques or the use of on-site field analytical kits (Schmidhalter,
2005; Yamin et al., 2020). However, these soil and water analyses are expensive and
time-consuming, which drives farmers and agricultural consultants to adopt over-
fertilization practices to maximize yield. Recently, spectral methods to analyze soil
nutrient content have also been developed (Zhang et al., 2016). However, nutrient
mobility in the soil, which is controlled by the water and fertilizer application
methodologies, along with the diurnal and seasonal root uptake, results in dynamic
fluctuations of the nutrient concentration across the soil profile (Dahan et al., 2014;
Turkeltaub et al., 2016). Often, nutrient concentration fluctuations in the soil range over
several orders of magnitude, from a very low concentration, considered nutrient
deficiency, to a very high concentration, considered excessive or even harmful to the
crop (Wolf et al., 2023; Wey et al., 2022). Therefore, optimal fertilizer adjustment that
accounts for the actual crop nutrient demand and avoids excessive fertilization cannot
rely solely on sporadic analyses of the soil nutrient state. Accordingly, optimization of
fertilizer application and prevention of water resource pollution require real-time
adjustments of fertilizer and water application that account for the actual variations in
soil nutrient concentration.
In this study, a novel soil nitrate monitoring system (SNS) (Yeshno et al., 2019), which
enables continuous in-situ measurement of the soil nitrate concentration, was
implemented in a full-scale field agricultural experiment. The study aimed primarily at
high-resolution characterization of the dynamic variations in soil nutrient concentration
across the soil profile in response to variations in fertigation pattern. Accordingly,
continuous data on variations in soil nitrate content enable frequent adjustment of the
fertigation regime in an attempt to achieve the desired nutrient concentration across
the soil profile. The SNS is based on continuous analysis of the soil porewater using



UV absorption spectroscopy combined with an algorithm for eliminating DOC interference with nitrate measurements (Yeshno et al., 2021). Although nitrate is one component out of nitrogen forms, it is an important indicator of the N-fertilizer state in the soil. In most agricultural soils (aerobic conditions), other forms of N-fertilizers (e.g., ammonium and organic nitrogen) eventually transform into nitrate through nitrification processes. Moreover, as opposed to other N forms, which tend to be absorbed by the soil, nitrate is a mobile form that is easily transported by percolating water and is, therefore, responsible for most environmental N pollution. Accordingly, the objective of this study is to enable fertilizer application adjustments during the growing season in an agricultural field based on real-time continuous measurement of soil nitrate concentrations across the soil profile. Ultimately, real-time adjustment of fertilizer application aims at achieving desired nitrate concentrations across the soil profile, while preserving optimal crop yield.

## 2. Materials and methods

### 2.1. Soil nitrate monitoring system

A custom-made soil nitrate monitoring system was constructed in Ben-Gurion University laboratories, in cooperation with DOTS Ltd. (Patent # US20200072737A1). The SNS enables real-time continuous monitoring of nitrate concentrations across the soil profile. The SNS's technical structure has been described in previous publications (Yeshno et al., 2019, 2021). Therefore, only a brief overview of the system structure is provided here. The SNS consists of a UV light source and a UV-VIS spectrometer that measures the absorbance properties of soil porewater within an optical flow cell. The flow cells are connected to customized suction cups that are installed at the desired depths across the soil profile. The SNS control panel contains a pumping system that generates a continuous low flux of soil porewater (< 10 ml/hour) through the optical flow cell. Each suction cup has its own optical flow cell for spectral analysis and real-time determination of the soil nitrate concentration across the soil profile. In addition, the SNS enables automated collection of the soil porewater for validation and calibration through lab chemical analysis. Customized suction cups, designed with a small dead volume and high sampling capacity, are connected to the optical flow cell through small-diameter tubing (1.6 mm inner diameter) to minimize the dead volume between the monitored zone in the soil and the optical flow cell in the control panel.





## 2.2. Study site

The experiment was conducted during the bell pepper growing season at the Yair Agricultural R&D Center, Central Arava Valley, Israel (30°46'40.1"N 35°14'21.8"E). The region is a hyper-arid desert with average annual precipitation of 28 mm and potential evaporation of 4,400 $mm*y^{-1}$ (Israel Meteorological Service, n.d.). Despite these harsh conditions, this region has been intensively cultivated for over six decades, using local groundwater combined with sophisticated floodwater harvesting systems and agricultural technologies. In recent years a growing component of desalinated water has been introduced to the valley water system. Unfortunately, intensive agriculture in the area has resulted in severe degradation of the groundwater quality, which is mainly reflected in elevated nitrate concentrations and salinity (Shalev et al., 2015).

## 2.3. Experimental setup

Bell peppers *(Capsicum annuum, Cannon* and *Galiano* varieties*)* were planted in a mesh greenhouse (30 m x 25 m (750 $m^2$)) on August 10, 2021, and the growing season lasted nine months till April 30, 2022. Twelve harvests were performed during the season, and the fruits were counted, weighed, and sorted according to their quality.

Nitrate concentration in the soil porewater was monitored by the SNS with an hourly resolution in 18 points, which were distributed in three depths (20, 40, 60 cm) in three replicates under two plots, experimental and control (Figure 1). In addition, water content sensors (GS3, Decagon devices) were installed adjacent to the suction cups in the experimental treatment. Bell pepper plants typically have a shallow root system where ~80% of the root length density is concentrated in the upper 30 cm of the soil profile, with almost no roots below 40 cm depth (Kong et al., 2012). Hence, nitrate measurements at depths of 20 and 40 cm are considered to represent the active part of the root zone for nutrient and water uptake, and nitrate presence at 60 cm is considered lost to down-leaching to the groundwater. The control treatment had a predetermined fertilization regime that relies on the standard practice growth protocol in the Arava region (Appendix 1). Fertigation of the experimental plot was frequently manipulated according to the observed variations in soil nitrate concentration. Due to operational delays, reliable SNS measurements started 100 days after plantation and



continued successfully for a period of five months, until the last harvest, 260 days after
plantation.

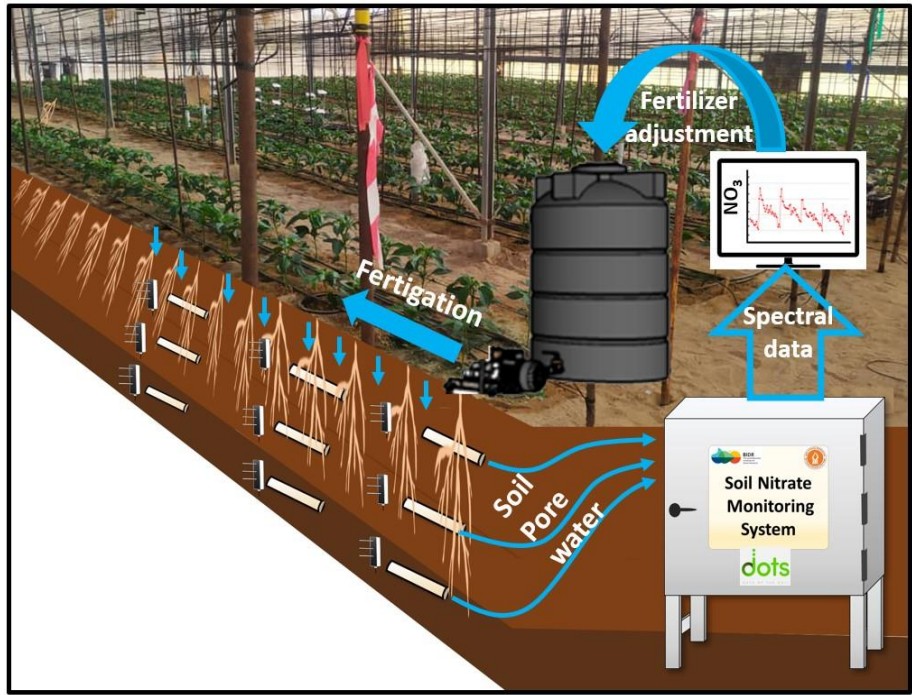

*Figure 1. Schematic illustration of the soil nitrate monitoring set-up and*
*fertigation adjustment process.*

The greenhouse consists of 40 plots, 12.5 m x 1.5 m each. The two treatments had
eight randomly distributed repetitions, while eight plots were used for margins, and
others were allocated to different experiments, which are not reported in this
manuscript. Each plot had two plant rows spaced 40 cm apart and with a 40 cm
distance between plants. Irrigation was conducted using a drip line (Netafim, 1.6 Lh$^{-1}$)
at 40 cm drip intervals along the line. Accordingly, the plant density was 3.3 plants per
1 m$^2$. The soil in the growing pad consisted of imported well-drained coarse sand.
Irrigation water for agriculture in the area mainly comes from brackish water from local
aquifers diluted with some desalinated drinking water, which yields water with an EC
of ~2–2.5 mS/cm. Fertigation was based on liquid fertilizer NPK 7-3-7 (Arava, ICL
group), in which 66% of the N is in nitrate form and 33.3% is ammonium. The fertilizer



was diluted in a tank and applied through the drip irrigation system by a fertilizer
injector. Both treatments were irrigated once or twice every day.
Throughout the first stages of the experiment, manipulations of the fertigation regime
in the experimental plot primarily aimed at investigating the dynamic variations in
nitrate concentration across the soil profile in response to variations in fertilizer and
water application. Later, real-time data on variations in nitrate concentration across
the soil profile were used to achieve the desired concentration range. In this
experiment, we defined a soil nitrate concentration threshold of 200 mg/L $NO_3$ as the
minimal desired value to prevent nutrient shortage and achieve a healthy crop.
Although nitrate concentrations of ~100–120 mg/L should be sufficient to avoid yield
loss (Kurtzman et al., 2021), here we deliberately aimed at higher threshold
concentrations since this experiment is pioneering and aimed at investigating the
ability to control the soil nutrient concentration and not necessarily to reach the lowest
possible concentration. Accordingly, fertilization adjustments were primarily made to
maintain the soil nitrate concentration at or above the threshold levels.
**2.4. Calibration and validation**
To measure nitrate concentration with UV absorption spectroscopy, a multiwavelength
method with a stepwise regression was implemented to overcome DOC interference
(Etheridge et al., 2014). Calibrating the nitrate concentration was carried out through
a set of soil porewater solutions that were collected from the field. The water samples
were collected from different points in the field and, therefore, contained a range of
DOC and nitrate concentrations. The initial DOC values of the soil water samples were
measured using an Analytic Jena multi-N/C 2100s TOC/TN analyzer, and nitrate
concentrations by Dionex ICS-5000 Ion Chromatography. To include a wide range of
nitrate levels, as one expects to find in agricultural fields, some of the samples were
spiked with $KNO_3$. Overall, 20 water samples of known nitrate and DOC
concentrations were taken, ranging from 5–25 ppm DOC and 90–750 ppm nitrate
($NO_3$). A stepwise regression was used to determine the linear combination of
wavelengths in the nitrate absorption range, which can predict the solution's known
concentration. Stepwise regression is a dimensionality reduction method that removes
less important predictors with an automatic iterative process. By the end of the



process, the stepwise algorithm yields a set of significant predictors, which have their
coefficient                                              as                                              follows:
$$y = a_1 x_1 a_2 x_2 \dots a_n x_n$$
Here, we identify a set of seven predictors that yield good results, and all the predictors
are statistically significant with a coefficient of determination ($R^2$) of 0.976 and a root
mean square error (RMSE) of 36.9. To validate the accuracy of the SNS spectral
measurements, water extracted by the SNS was sampled approximately once every
two weeks and analyzed through the standard laboratory method. The samples were
filtered through a 0.45-micrometer filter and stored refrigerated at a temperature of 4
°C. Over the entire period (160 days), 60 samples were randomly selected (~20
samples for each depth from three different locations) for nitrate lab analysis using
Dionex ICS-5000 Ion Chromatography. Comparing the nitrate concentration
measurements that were continuously taken in the field, by the SNS, with the nitrate
concentration measurements of the water samples that were frequently collected
indicated high accuracy and reliability ($R^2$ of 0.916 and an RMSE of 50.97) (Figure 2).

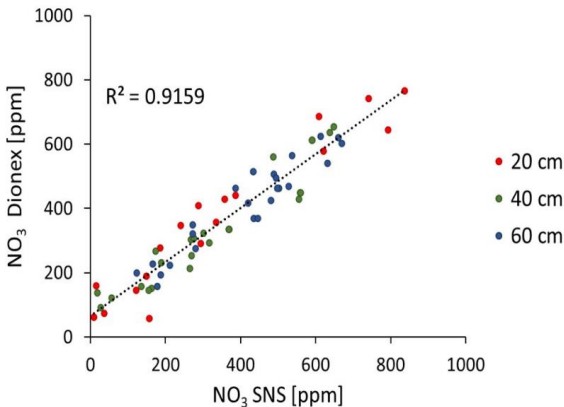

*Figure 2. Nitrate concentration measurements by the SNS*
*vs. standard laboratory analysis.*

It should be noted that while the SNS measured the nitrate concentration online in the
flow cell, the water sample that was used for validation was a cumulative sample that
was collected over several hours in a sampling cell. In light of the diurnal variation in
concentration (see Results section on diurnal variations in soil nitrate concentration),



it is obvious that some differences between the online SNS measurement and the
cumulatively collected water sample are expected.

## 3. Results and discussion

### 3.1. Controlling soil nitrate concentrations

In the following, we present the dynamic variations of nitrate concentration and water
content across the experimental plot's soil profile in response to variations in fertilizer
and water application (Figure 3). Soil nitrate concentration is presented as the daily
average of the hourly measurements at the three points for each depth (Figure 3a).
The soil water content is presented as the average values of three spatial points in an
hourly resolution (Figure 3b).

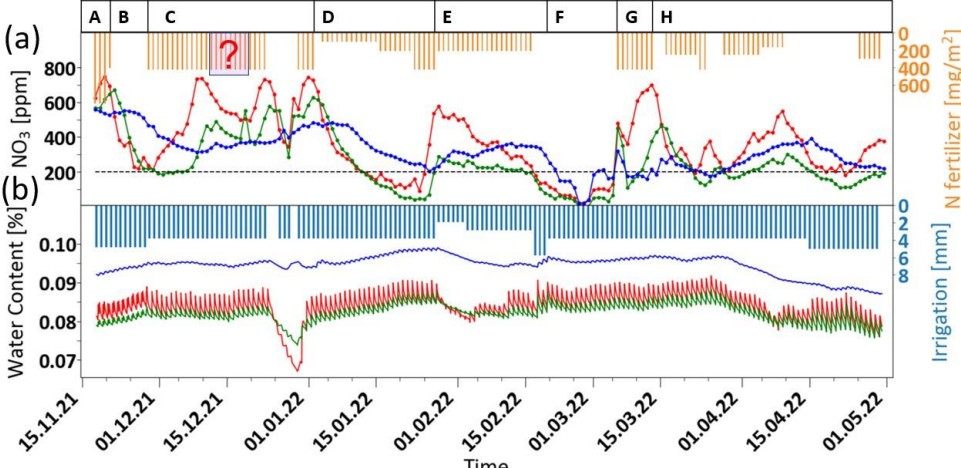

*Figure 3. Variations in soil nitrate concentration (a) and water content (b) in the experimental plot during the growing season, along with the daily irrigation and N-fertilizer application.*

The variations in both nitrate concentration and soil moisture are presented, along with
the irrigation and fertilization application quantities. During the first 100 days of the
growing season, the water and fertilizer application in both plots was mostly fixed, with
approximately 4 mm*day$^{-1}$ water and 425 mg*m$^{-2}$*day$^{-1}$ N-fertilizer (equivalent to 106
ppm of nitrate in the irrigation water). For convenience, variations in the irrigation and



fertilizer amounts will be described here as a percentage of the fixed amount that is
the standard practice in the region (Appendix 1).
As soon as the nitrate measurements began (100 days after plantation), nitrate
concentrations across the entire soil profile exhibited extremely high values, ranging
from 625 to 750 ppm (stage A in Figure 3a). This concentration range is ~6 times
higher than the maximum concentration viable for plant uptake (Kurtzman et al., 2021).
The daily fluctuations in soil water content across the profile provide a clear indication
that during this stage, a substantial flux of nitrate leaches down from the root zone to
deep sections of the unsaturated zone and ultimately to the groundwater (stage A in
Figure 3b).
Following the observation of the very high soil nitrate concentration under the common
fertigation program, fertilizer application was completely halted under the experimental
plot for a period of eight days (stage B in Figure 2a). During this time, water application
was maintained at 100% of the prescribed dose. As a result, the nitrate concentrations
at depths of 20 and 40 cm dropped dramatically to the threshold concentration (~200
ppm) at a rate of ~180 ppm per day. At a depth of 60 cm, a gradual decrease in nitrate
concentration only started five days later, reaching a minimum concentration value of
290 ppm 13 days later, at a rate of ~50 ppm per day. Obviously, the observed reduction
in nitrate concentration may be attributed to both plant consumption and to transport
and down-leaching. After the nitrate concentration in the root zone reached the
threshold values, fertigation resumed at 100%, and the nitrate concentration at 20 cm
immediately rose to 750 ppm (stage C in Figure 3a). Deeper in the soil profile at a
depth of 40 cm, the nitrate concentration increase was delayed for two days to ~450
ppm. During this period, the nitrate concentration below the root zone at a depth of 60
cm remained stable at ~300 ppm. This stage enabled defining the time lag that is
required to achieve a significant decrease in nitrate concentration in the profile,
following a reduction in fertilizer application, and the recovery time after fertilizer re-
application. Moreover, the concentration differences between the depths reflect both
the root uptake and the unavoidable nitrate down-leaching to the unsaturated zone
under these conditions. Note that during stage C, there were two periods with
unexpected decreases in nitrate concentration, which are especially notable at the 20-
cm depth. Both are obviously a direct indication of a failure in fertilizer application.
While the first period was unnoticed and not recorded (marked by a question mark on



the fertilizer application bars), the second reduction in nitrate concentration raised an alarm for a system check, and indeed, a break in fertilizer application was detected and fixed. As a result, the nitrate concentration in the soil rose again to the max value of 750 ppm at 20 cm and 600 ppm at 40 cm.

Since, during stage C, the concentration in the soil rose again to undesirably high levels, an adaptive fertilization regime was implemented (stage D in Figure 3). In this stage, the fertilizer application was evaluated according to the actual measured soil nitrate concentration, and the fertilizer amount was frequently changed, while the irrigation amount remained fixed. Reducing the fertilizer amount to 25% resulted in an immediate and sharp concentration drop toward the threshold. To avoid a nitrate concentration decrease below the threshold values, fertilizer application was increased to 50%. However, the decreasing trend did not stop, and a few days later, the concentration dropped below the threshold. Hence, 100% fertilizer amounts were implemented again, and the nitrate concentration at 20 cm quickly rose to undesirably high values (570 ppm). Throughout this period, a gradual decrease in nitrate concentration, from ~400 to ~200 ppm, was also observed at 60 cm, providing encouraging indications of a reduction in nitrogen flux from the root zone down to the deep unsaturated zone.

Even though stage D provided significant potential to control the nitrate concentration in the root zone, the concentration below the root zone was still high, reaching ~200 ppm, which is far above the requirement for safe groundwater recharge (Directive, 1991). However, this stage was conducted with a fixed amount of irrigation water and variable fertilizer applications. To reduce nitrate leaching below the root zone, the fertilizer dose was reduced to a fixed value of 50%, while the irrigation amount was changed in an attempt to reduce nitrate transport below the root zone (stage E in Figure 3). It has been hypothesized that since the soil is composed of coarse sand, excessive irrigation leads to the leaching of water with a high nitrate concentration below the root zone before the plant uptake is completed. Accordingly, it has been hypothesized that reduced water application could increase nitrate retention in the root zone. During the first six days of stage E, the irrigation was reduced to 50%. As a result, the nitrate concentration at 20 cm gradually dropped, while at 40 cm, the concentration remained near the threshold values. As expected, the reduction in water



application resulted in a decrease in water content at all depths (stage E in Figure 3b). However, water shortage in the soil resulted in increased salinity that was noticed on the plant as leaf damage. Hence, irrigation was increased to 75% for eight days and later to 100% for an additional six days. Soil salinity was enhanced since this field is irrigated with high-EC brackish water (2–2.5 mS/cm). As such, preventing salinity impacts requires a high leaching fraction to avoid salt accumulation. To prevent salinity damage to the crop, the irrigation was increased to 150% for three days to wash away the accumulated salts and allow plant recovery.

During stage F (18 days), no fertilizer was applied due to a technical failure in the fertilization system. However, water application was maintained at a 100% level. Obviously, this resulted in soil wash down, which was reflected in a dramatic decrease in nitrate concentrations at all depths to practically zero concentration, which is obviously below the desired threshold and dangerous to the crop. Accordingly, in stage G, fertilizer application was intentionally increased again to 100%, although it is obvious that the concentration in the soil would rise again beyond the desired levels. At this stage, no attempt to maintain nitrate levels at the concentration threshold was made, to allow other research groups in the project to examine the relation between soil nitrate concentration increase and nitrous oxide emissions. This part is beyond the scope of this paper.

To reduce the magnitude of fluctuations around the threshold, it has been concluded that daily evaluation of variations in the concentration trend is necessary for proper decision-making on the fertigation regime (stage H in Figure 3). In this stage, the irrigation was kept constant at 100%, while the fertilizer was adjusted daily according to the measured variations in nitrate concentration across the soil profile. During the last 50 days of the growing season, one out of four fertilization regimes was selected for application, 100%, 50%, 25%, or 0%. It should be noted that this approach was conservatively biased to satisfy the plant needs and prevent concentration drops below the threshold value while considering that a change in the fertilization regime can affect the soil in a lag-time of 2–3 days. This fertigation approach successfully decreased the nitrate concentration in the root zone and partially stabilized the concentrations around the desired threshold. Accordingly, it may be concluded that frequent adjustment of fertilizer application can improve the ability to control the



fertilization regime at the desired threshold. However, the nitrate concentration at 60
cm, which represents the leachate concentration, was still very high, 200–400 mg/l,
similar to the concentration in the root zone. Obviously, this is a direct result of the low
irrigation efficiency, which is dictated by the irrigation water's high salinity.
Nevertheless, during stage H, when the nitrate concentration in the root zone was
preserved slightly above the desired threshold, the fertilizer application was only 52%
of the recommended amount.
Soil nitrate measurements under the control plot show that for most of the season, the
soil nitrate concentration remained significantly high, ranging approximately from ~500
to ~800 ppm at all depths (Figure 4). Note that fertigation of this plot was presumably
fixed on daily rates, which were prescheduled according to the recommendation for
growers in this region (Appendix 1).

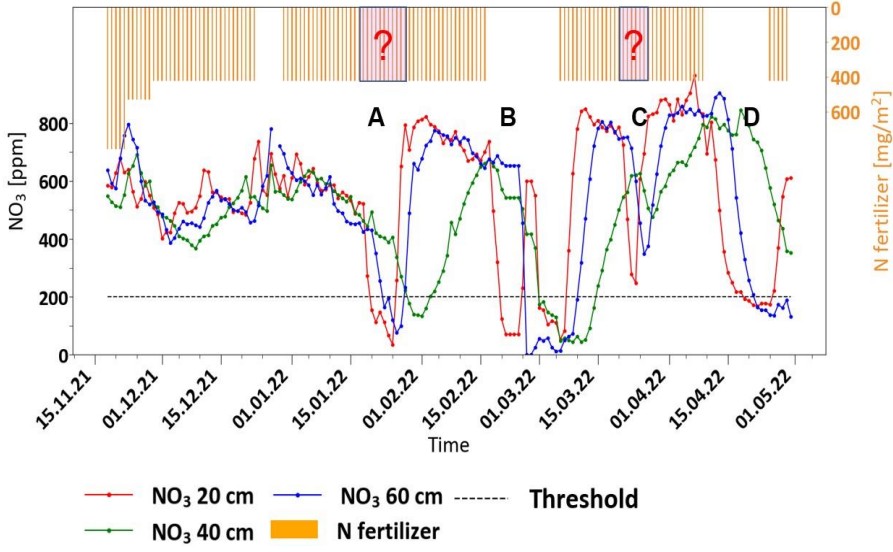

*Figure 4. Variations in soil nitrate concentration and daily N-fertilizer application in the control treatment.*

Although the data from the control plot were not used for managing fertigation, the
recorded variations in nitrate concentration exhibit four distinct periods in which a
sharp decrease in nitrate concentration was monitored (stages A, B, C, and D).



Obviously, such a reduction in soil nitrate concentration is a direct outcome of a technical failure in the fertigation system, which was supposed to provide daily fixed fertigation amounts. In two of these events (stages B and D in Figure 4), the reduction in soil nitrate concentration was noticeable and also recorded as a technical failure in the fertigation system. On the other hand, in the two other events (marked as A and C), the fertilization failure was not detected and, therefore, not indicated as a no-fertilization period. Nevertheless, the sharp reduction in soil nitrate concentration provides a retrospective identification of a fertigation problem, which may raise an alarm for an actual fertigation problem and potential nutrient deficiencies to the crop. The measured high nitrate concentration across the entire profile, during most of the growing season, reflects excessive fertilization with evident down-leaching of nitrate from the root zone to the groundwater. Obviously, these results show that predetermined fertigation schedules that do not take into account the actual soil nutrient state lead to excess nutrients in the root zone, as well as groundwater pollution.

## 3.2. Diurnal variation in soil nitrate concentration

Throughout the experiment, decision-making on fertigation management relied on daily averages of hourly nitrate measurements (Figures 3). Nevertheless, the hourly resolution reveals a notable diurnal variation in soil nitrate concentrations, from ~400 to ~900 ppm for the presented period of 12 days in Figure 5. The variations in soil nitrate concentration follow fertigation event patterns, along with the diurnal crop nutrient consumption. Close inspection of variations in soil nitrate concentration shows a sharp concentration increase immediately after each morning fertigation, which is followed by gradual decrease during daytime. The daily fluctuations in concentration were particularly significant with the 100% dose fertigation (500–600 ppm), and slightly smaller with a 50% dose (300–400 ppm). Obviously, the sharp increase in nitrate concentration is attributed to the infiltration of a concentrated irrigation solution. However, the nitrate concentration in the irrigation water is only 240 ppm $NO_3$. Therefore, the top daily concentration of 700—900 ppm reflects down-leaching of the concentrated solution from the top 20 cm of the soil profile, where evapotranspiration is most effective. On the other hand, the daily decrease in nitrate concentration is directly attributed to nitrate consumption by the roots, which is equivalent to ~50 ppm





$m^2$ day. Although nitrate consumption may be the main reason for the observed daily
decrease in nitrate concentration, it is possible that other microbial processes, such
as denitrification and N-oxide release, could also contribute to reductions in nitrate
concentration. These results highlight the potential of micro-managing fertilizer
application to reduce nitrate leaching and to enhance nutrient uptake by plants.

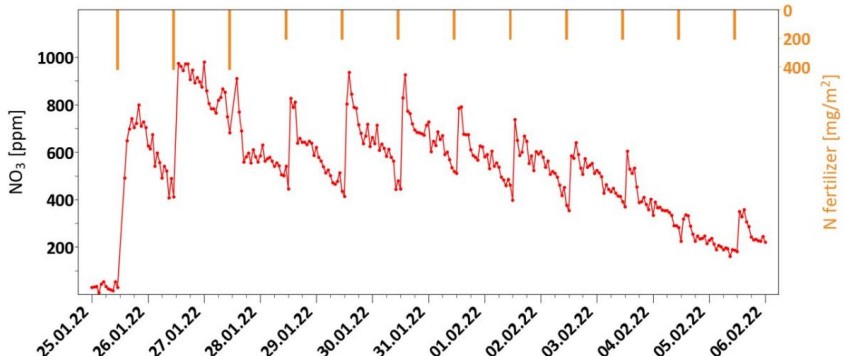

*Figure 5. Soil nitrate concentration and daily N-fertilizer dose in the control treatment.*

### 3.3. Fertilizer application and yield

During the monitoring period (60% of the growing season), 559 kg N/hectare was
applied in the control treatment and only 349 kg N*hectare$^{-1}$ in the experimental plot,
exhibiting a 38% reduction in fertilizer application. Nevertheless, reducing N-fertilizers
did not affect the total yield, which included 12 harvests during the season. The total
yield in the control plot reached 95.6 ton*hectare$^{-1}$, while the total yield in the
experimental plot was slightly higher, reaching 105.6 ton*hectare$^{-1}$. This difference
was found to be non-significant in a t-test (p-value = 0.35, degree of freedom = 14).
Furthermore, fruit quality was also slightly higher, although also non-significant, in the
experimental plot where 48.4% of the total fruits were classified as export yield, with
47.6% classified as such in the control plot. The export yield was 46.0 tons per hectare
in the experimental plot and 50.4 tons per hectare in the control plot (p-value = 0.374,
degree of freedom = 14). A reduction in fertilizer application was achieved even though
the soil nitrate concentration was mostly maintained above the 200-ppm threshold.



These results emphasize that fertilizer application that is carried out through a fixed
protocol, as is commonly practiced today in most agricultural fields all over the world,
releases high nitrate concentrations into the environment, which are three to five times
higher than the level required for a healthy crop and optimal yield.

## 3.4. Nitrate down-leaching

To calculate nitrate down-leaching, we considered the leachate fraction of the irrigation
water and the nitrate concentration at a depth of 60 cm. Previous studies on bell
pepper crops in this region have shown that this area's high-salinity water requires
irrigation amounts that are at least twice the actual plant evapotranspiration to prevent
salt accumulation in the soil (Ben-Gal et al., 2008). Plant daily evapotranspiration for
this crop in this region at different growth stages had previously been calculated in a
large-scale lysimeter experiment conducted by the R&D Center (Water recycling
project, Pharan 2007, unpublished, in Hebrew), and irrigation recommendation tables
for the farmer were published, including daily evapotranspiration and recommended
irrigation amounts (Friedman et al., 2022). Following these recommendations,
irrigation was carried out in both the experimental and the control plots for most of the
season. The calculation of the cumulative nitrate flux is performed as follows:

$$cumulative\ NO_3\ flux = \sum (I - et) * NO_{3(60\ cm)}$$

where $I$ is the daily irrigation amount, $et$ is the daily actual evapotranspiration, and
$NO_{3(60\ cm)}$ is the daily average of nitrate concentration at 60-cm depth.
Cumulative nitrate leachate amounts for the monitored period were 1242 and 872 kg
$NO_3$/hectare in the control and experimental plots, respectively. This significant
reduction of 30% in nitrate down-leaching was achieved even though fertilizer
adjustments were performed manually for part of the growing season. Although nitrate
down-leaching was successfully reduced in the experimental plot, the results still
indicate extremely high nitrate leachate. Furthermore, when considering the fraction
of N loss through leachate in relation to the applied total N-fertilizer, it was found to be
similar in both the experimental and the control treatments, with values of 56% and
50%, respectively. This suggests that nitrogen efficiency was very low, and a
significant portion of the N-fertilizer is transformed into nitrate, which then leaches



down from the root zone to the groundwater. Since nitrate leachate is influenced by
both water flux from excessive irrigation and nitrate concentrations below the root
zone, both water efficiency and nitrogen efficiency play crucial roles in minimizing N
leaching. However, it is essential to note that the results presented here are based on
an experiment conducted in very coarse sandy soil, irrigated with high-salinity water.
Under such conditions, high drainage and difficulty in controlling nitrate down-leaching
were observed in both treatments. It can be noted that during the last month of the
growing season, applied irrigation closely matched actual evapotranspiration.
Consequently, there was hardly any nitrate leaching, although the nitrate
concentration increased below the root zone.
**3.5. Fertistat**
During the experiment, real-time measurements were used to guide fertilization
decisions, but the exact amounts of fertilizer were still determined through trial and
error. This method resulted in a 38% reduction in fertilizer usage but could be further
optimized with the use of an algorithm that calculates required fertilization amounts
based on real-time measurements. Such a fertigation algorithm, based on continuous
analysis of soil nitrate, would function as a "fertistat" that controls the soil nitrate
concentration at the desired levels. Analogous to temperature control by a thermostat
in any heat process, a fertistat mechanism would act to achieve the desired soil
nutrient concentrations that are required for optimal yield, while preventing nutrient
excesses or deficiencies. Hence, the fertistat mechanism does not depend on soil
type, plant demand, or climatic conditions, as it enables direct control of the soil
nutrient concentration to attain these desired values. Results from this study show that
application of a fertistat mechanism may dramatically reduce fertilizer application while
achieving high crop yield.
**4. Conclusion**
Continuous in-situ monitoring of nitrate concentrations across the soil profile was used
for real-time management of fertilizer and water application in an agricultural field. The
following conclusive results were obtained throughout the experiment.
1. A soil nitrate monitoring system (SNS), which is based on continuous spectral
analysis of the soil porewater, was operated during the growing season of a bell
pepper crop in a greenhouse. The SNS exhibited robustness and accuracy, which



   proved its suitability for optimizing fertilizer and water application in agricultural field conditions.

2. Real-time monitoring of the soil nitrate concentration revealed the dynamic responses of the soil to water and fertilizer application. Hourly measurements showed daily fluctuations in nitrate concentrations, which correspond well with the daily fertigation events and plant nutrient demand.

3. Soil nitrate concentration under the control plots, which were fertigated according to the standard regional fertigation plan, exhibited a very high concentration range of ~600 to ~800 ppm, which persisted for most of the growing season (apart from short periods of technical failure in the fertigation system). Since achieving a healthy yield of bell pepper, in this particular case, requires soil nitrate concentrations of 80–150 ppm, these results demonstrate that fertigation that is based on a prescheduled fertigation plan can lead to excessive fertilization, posing the risk of severe groundwater pollution.

4. Continuous data on variations in soil nitrate concentrations enabled manipulating the applied fertigation regime while driving the soil nitrate towards the desired concentration range. In this experiment, the nitrate concentration threshold was set at 200 ppm, which is well above the minimum required for max yield. Nevertheless, a reduction of 38% in fertilizer application was achieved without affecting the total yield or the fruit quality.

5. Manipulation of the fertigation plan to achieve the desired soil nitrate concentration resulted in a reduction of 30% in nitrate flux below the root zone. This poor result, which did not meet the aim of reaching zero down-leaching, is attributed to very low irrigation efficiency, which is dictated by the irrigation water's high salinity (EC of ~2–2.5 mS/cm). Therefore, additional reduction of nitrate down-leaching requires irrigation with lower salinity water.

6. Combining real-time soil nitrate monitoring technology with an automated fertigation program has the potential to significantly reduce fertilizer usage, eliminate nitrate down-leaching, and prevent water resource pollution.



## ACKNOWLEDGEMENTS

This research was funded by the: (1) The Israel Ministry of Agriculture and Rural Developmentunder the project name "Reducing nitrate fluxes to groundwater from agricultural fields," as part of the program The Root of the Matter, and (2) the Nekudat Hen Foundation. In addition, scholarships were granted by (1) the Harbour Foundation for a Fay and Bert Harbour Award, (2) the Yair Guron Scholarship Fund, and (3) the JNF Scholarship Program. We would like to express our gratitude to Michael Kugel and Yuval Shani for supporting all the technical aspects of this research.

## AUTHOR CONTRIBUTIONS

Conceptualization: Y.Y, O.D.; experimental setup: Y.Y., Y.R., O.D.; analysis: Y.Y., Y.R., O.D.; writing: Y.Y., O.D.; project supervision: O.D., S.A.

## COMPETING INTERESTS

The authors declare no competing interests.

## DATA AVAILABILITY

All data generated during the current study are available from the corresponding author on reasonable request.



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

| Days after planting | Plant development stage | N concentration in irrigation water (ppm) | Desired $NO_3$ concentration in soil |
|---|---|---|---|
| 0–30 | Growth | 50–70 | 250 |
| 30–45 | Early fruit set | 50 | 0–50 |
| 45–50 | Late fruit set | 120–150 | 300–400 |
| 50–125 | First harvest | 100–120 | 250–300 |
| 125–165 | Winter harvest | According to soil test results | 250 |
| 165–270 | Spring harvest | According to soil test results | 250 |

