# Peer review of "Optimized fertilization using online soil nitrate data"

_EGUsphere, 2023_

## Referee Comment (RC1)

A review of: *Optimized fertilization using online  soil nitrate data*

By Yekutiel et al.

Summary and Recommendation

The manuscript describes a pioneering field trial of a new, relatively accurate, soil-nitrate-sensor (SNS) that can measure nitrate concentration in the soil porewater continuously. Soil nitrate was monitored at 3 depths (20, 40, 60 cm) under plots of 2 treatments: 1) prescheduled irrigation and nitrogen (N) fertigation recommended for intensive growing of bell-pepper in greenhouse in the arid Arava Valley (including saline irrigation water and course texture soil); 2) fertigation (and irrigation) was adjusted according to the online porewater-nitrate readings in a trial and error fashion (no controlled algorithm). Yield was non-significantly higher in the adjusted treatment # 2, nevertheless, N application was 38% smaller and nitrate deep leaching 30% smaller in the adjusted treatment 2. These results are promising for future controlled monitored algorithms of N application which will reduce the environmental impacts of intensive agriculture.

The description of the monitored data of the SNS and the potential agricultural and environmental positive impact that such controlled systems can achieve makes the manuscript very attractive for growers, water resources managers, soil scientists etc. The pre-matureness of the controlled field application of the system described here (no algorithm) does not reduce much for this pioneering work. Therefore, I recommend publication following moderate revisions suggested herein.

Major Comments

1) It is much easier for agricultural related nitrogen discussions to use the N, NO3-N, NH4-N concentration convention rather than the NO3 molecule. I recommend to use the 4.43 factor for nitrate and convert all nitrogen species concentrations to N concentrations.
2) The terminology of "eliminating" nitrate leaching or groundwater pollution is inappropriate in any setup of intensive agriculture on earth's soil (rather than soilless media). We can reduce and even reduce tremendously but not eliminate leaching.

Specific Comments

1) P2L4, "eliminate" see major comment 2
2) P3L20 see also Levy et al., 2017, Hydrol. Earth Syst. Sci., 21, 3811–3825
3) P6L5 potential evaporation of 4400 mm/yr sounds too high, more in the area of 2000-2500 from what I know, check.
4) P9L1-4, I assume the multiple regression results in a predicted nitrate concentration (Y) in the form of: Y = a1x1 + a2x2 + a3x3 … (no + signs in the text).
   It would be much better to write the regression model explicitly. What are the 7 adsorption wavelengths predictors? and their coefficients.

5) P10L8, delete" at the three points"
6) Figure 3a – missing legend, what color is what depth?
7) P10L15-16, 106 mg/l nitrate or 106 mg/l N? check
8) P15L32 "equivalent to ~50 ppm m2 day" unclear, perhaps 50 mg/m2/d?
9) Figure 5, concentration in what depth?
10) P16L11-20. I assume the total yield reported is the mean of 8 plots in each treatment (T test answers the question: is the mean of the 2 populations (replicas of the each treatment here) is significantly different?). A table with the statistics of all yield variables (total, high quality, etc) and leachate including (# of replicas, mean, StD, P(Tsest) of each pair (control, experiment) would be a much better presentation of these results.
11) P17L21-30, show the difference in leachate is significant (Ttest), and non significant difference for leaching fraction. Consider all in 1 table together with yield results as suggested in comment # 10.

---

## Author Response (AR1)

**Reply to reviewer comments**

We wish to thank the reviewers for their encouraging review. We have addressed all comments and added clarification according to the questions. We acknowledge the review process and appreciate very much the suggested improvement. The following are our replies to the reviewer's comments.

**Revier 1  04-01-2024**

**Major Comments**

1. It is much easier for agricultural related nitrogen discussions to use the N, $NO_3$-N, $NH_4$-N concentration convention rather than the $NO_3$ molecule. I recommend to use the 4.43 factor for nitrate and convert all nitrogen species concentrations to N concentrations.
   Comment accepted. The nitrate values were changed both in figures and text. While the term 'nitrate concentration' will still be used in the text, when providing quantitative values, we will explicitly state that it refers to N-$NO_3$.

2. The terminology of "eliminating" nitrate leaching or groundwater pollution is inappropriate in any setup of intensive agriculture on earth's soil (rather than soilless media). We can reduce and even reduce tremendously but not eliminate leaching.
   Comment Accepted. The terminology has been adjusted to reflect a more achievable and realistic objective .(P2L3, P21L8)

**Specific Comments**

1. P2L4, "eliminate" see major comment 2
   Comment accepted (P2L3)
2. P3L20 see also Levy et , 2017, Hydrol. Earth Syst. Sci., 21, 3811–3825
   Comment accepted. Citations were updated (P3L21)
3. P6L5 potential evaporation of 4400 mm/yr sounds too high, more in the area of 2000-2500 from what I know, check.
   The provided data is based on measurements from the Israel Meteorological Service of daily evaporation from a bucket for the years 1994-2013 in Hatzeva (a meteorological station near the experiment site) [https://ims.gov.il/en/data_gov]. Literature data for the Arava region suggests a potential evaporation of 3500 mm/year, but this information is available only from the year 2001, and approximated for the whole region and therefore, less accurate. [Goldreich, Yair, and Ora Karni. "Climate and precipitation regime in the Arava Valley, Israel." Israel journal of earth sciences 50 (2001).]
4. P9L1-4, I assume the multiple regression results in a predicted nitrate concentration (Y) in the form of:  $Y = a_1x_1 + a_2x_2 + a_3x_3 \ldots$ (no + signs in the text).

Comment accepted. The intended result of the multiple regression includes only linear terms without interaction between predictors, the equation has been corrected accordingly (P9L3)

5. It would be much better to write the regression model explicitly. What are the 7 adsorption wavelengths predictors? and their coefficients.

Comment accepted and a table with relevant wavelengths and coefficient was added to the manuscript  (P9L8)

6. P10L8:" Soil nitrate concentration is presented as the daily average of the hourly measurements at the three points for each depth". delete" at the three points"

The average values represent data from hourly measurements collected from three different points for each depth which were aggregated to daily resolution for each depth. This is described in more detail in the method chapter (P6L20-23). To enhance clarity, we revised to " Soil nitrate concentration is presented as the daily average of the hourly measurements at the three different measuring points for each depth" (P10L6)

7. Figure 3a – missing legend, what color is what depth?

Comment accepted. The legend for Figure 3a was mistakenly truncated during submission, this has been solved and the legend is properly displayed.

8. P10L15-16, 106 mg/l nitrate or 106 mg/l N? check

Comment accepted. It indeed refers to 106 mg/l N. The manuscript was corrected accordingly. (P10L12-13)

9. P15L32 "equivalent to ~50 ppm m2 day" unclear, perhaps 50 mg/m2/d?

The expression "equivalent to ~50 ppm m2 day" was included in the text inadvertently and is not essential for text clarity or understanding of the diurnal variations of soil nitrate concentration. The expression was omitted. However, this error provides an opportunity to revisit this paragraph and the suggested process of the diurnal concentration reduction. As a result, the paragraph has been rephrased, with an emphasis on the downward flow of irrigation water along with transport, plant uptake and evaporation processes (P16L20-26).

10. Figure 5, concentration in what depth?

This figure refers to 20 cm depth. The caption and legend were corrected accordingly (P17)

11. P16L11-20. I assume the total yield reported is the mean of 8 plots in each treatment,  unclear from text (T test answers the question: is the mean of the 2 populations (replicas ) significantly different?). A table with the statistics of all yield variables (total, high quality, etc) and leachate including # of replicas, mean, StD, P(Tsest) of each pair (control, experiment) would be a much better presentation of these results.

The comment is accepted. The yield reported in the study represents the mean of 8 plots in each treatment. The manuscript was revised accordingly and a table presenting the descriptive statistics and t-test results for all yield variables was added (P18L3-4).

12. P17L21-30, show the difference in leachate is significant (Ttest), and non significant difference for leaching fraction. Consider all in 1 table together with yield results as suggested in comment # 10.

The comparison of leachate between the two treatments is primarily qualitative as the flux beneath the root zone cannot be directly measured but only estimated. Therefore, we believe that statistical tests may have limited relevance. The manuscript's main focus is on the ability to monitor and control nitrate concentration in the soil, with the leaching serving as an initial assessment of the potential for reducing nitrate pollution. We plan to conduct future experiments that will further investigate the leaching fraction and nitrogen leaching.

**Reviwer 2 – 31/01/2024**

**Major Comments**

1. Spatial variability at microplot level: how is taken into account by the suction cups system? How many suction cups per "reading"? Spatial variability does impact the analysis of data obtained from a point sensor. While the experiment assumes relatively homogeneous soil (imported sandy soil), each depth and treatment includes three suction cups located at different points ~1 meter from each other under the same growing raw to achieve three repetitions for each depth. Each suction cup is connected to a different flow cell where the optical measurements are conducted (see P5L24-28 and P6L20-22). Nevertheless, it should be noted that the suction cups used here are rather large (12 cm long; 2.5 cm OD; <4 cc dead volume), which enables large soil sampling space that inherently overcomes some of the local heterogeneity.

2. Soil characteristics: for understanding for readers that are not familiar sandy soils and arid climate a short description of the main soil characteristics would be useful e.g. texture, pH, Corg% and I would also be interested in knowing the homogeneity of the soil in the study area if you have information about that. E.g. the pad was filled with imported soil? Comment accepted. A brief description of soil characteristics, including texture, pH, and organic carbon percentage has been incorporated. (P7L11-12). This information is derived from soil samples taken at four random points within the greenhouse, at a depth of 0-30 cm.

Figure 3. This is a complex and key figure in the paper, I believe a reader should be able to understand the figure with the help of the caption without the need for the text. Therefore, please add a legend for colour of the lines as in Figure 4 and adapt the caption explaining the meaning of the letters (periods) and the colour of the three different lines (profile depths). At least in the PDF that I have it is not visible, but It looks like the figure is cut. You should also indicate more clearly that this plot refers to the "adapted" fertigation treatment only or I suggest considering merging Fig 3 and 4 together for a more direct visual comparison.

The legend for Figure 3 was unintentionally truncated during submission. The figure was corrected to ensure proper display. In addition the caption was updated to include explanations of the periods which are marked by letters to explicitly it refers to the 'adapted' treatment. Note that the term "adapted" was replaced throughout the text by "adjusted fertigation". We agree with the comment that Figure 3 is a key figure in the manuscript introducing the first instance the concept of adjusted fertigation that enables controlling soil nitrate concentration. Nevertheless, we believe that this figure should stand alone and not be merged with Figure 4.

3. How is the threshold value of 200 ppm decided? Is it crop specific and experience based?
   The threshold value of 200 ppm was determined based on research that investigated various crops and identified a maximum concentration above which plants cease to increase nutrient uptake ( see Kurtzman et al., 2021). Different crops exhibit different threshold values, generally falling within the range of 80-200 ppm $NO_3$. Since there wasn't a specific value available for bell peppers, and to ensure no damage to the crop, we chose a conservative threshold of 200 ppm. Note that $NO_3$ values in the revised manuscript were replaced with N-$NO_3$ values. Accordingly, the threshold value in the manuscript is now 45 ppm N-$No_3$ (See P8L8-15).

4. it looks that in the 40 cm layer the concentration took longer to increase after the downwards peak, why do you think it was the case?
   The delayed response of increase and decrease in nitrate concentration at depth of 40 cm compared to 20 cm is directly attributed to transport and uptake processes. 40 cm depth is the lower boundary of the root zone. Accordingly, the lag time is influenced by both reduced water and nitrogen fluxes in that zone compared to 20 cm.

5. Fig 5. Very interesting to see the substantial daily variations. However how come the N fertilizer rate changes from 400 to 200 mg / m² after the 29.01.? Isn't this the control rate which in Fig. 4 appear to have constant rate in this period? Although it is close to the question mark / error. From which profile are the data shown?
   Figure 5 presents the adjusted fertigation plot where the fertilizer was adjusted frequently, as presented in figure 3, and not from the control plot which was fertigated on a prescheduled program (except for the technical failure marked with question marks). These data correspond to the shift from the end of period D (100% fertigation) to period E (50% fertigation) in Figure The paragraph had rephrased to include this information. This time interval presents the change in fertigation regime and its impact on the magnitude of daily fluctuations (P16L3, P16L10-14).

6. Conclusion 5. Zero down-leaching is a far too optimistic objective in most systems, in sandy soil I guess especially (salinity constraint). So I would not define a reduction of 30% in nitrate flux as a poor result but rather put it in perspective with the ground water quality

objective.

Comment accepted. The sentence has been revised to offer a more balanced perspective, acknowledging both the achievements in nitrate reduction and the ongoing challenge of minimizing nitrate leaching to meet groundwater quality objectives.

**Specific Comments**

1. P1L27: instead of "dramatic" consider substantial or similar (less "dramatic") synonym
   Comment accepted, the term "dramatic" has been changed to "substantial" (P1L26)
   P6L21: usually city and country of the producer are indicated for a device e.g. Decagon devices appears now to be Metere Group, inc., Pullman, WA, U.S.A.)
   Comment accepted, the producer name has been changed as suggested. (P6L23)
2. P11 L13: Figure 3a?
3. Figure 3 consists of two axes: the upper one is labeled 3a, and the lower one is labeled 3b, as indicated by the lower case labels near the axes.
4. P16L3: "Nitrous oxide" if you are referring to the gas product of denitrification ($N_2O$)
   Comment accepted, The meaning will be clarified by adjusting it to "Nitrous oxide ($N_2O$)". (P14L2)
5. P16L8+.. You can abbreviate hectares to "ha"
   Comment accepted.

**Editor notifications:**

1. With the next revision, please add a caption to the appendix`es table according to our standards: https://www.soil-journal.net/submission.html#manuscriptcomposition > 7. Appendices.
Comment accepted

2. Regarding your figure 3: please ensure that the colour schemes used in your maps and charts allow readers with colour vision deficiencies to correctly interpret your findings. Please check your figures using the Coblis – Color Blindness Simulator (https://www.color-blindness.com/coblis-color-blindness-simulator/) and revise the colour schemes accordingly.
Comment accepted. color schemes had adjusted and unique symbols were added to allow readers with color vision deficiencies to interpret the data.